# Inherent Tradeoffs in Learning Fair Representations

**Han Zhao**[*]
Machine Learning Department
School of Computer Science
Carnegie Mellon University
`han.zhao@cs.cmu.edu`

**Geoffrey J. Gordon**
Microsoft Research, Montreal
Machine Learning Department
Carnegie Mellon University
`geoff.gordon@microsoft.com`

## Abstract

With the prevalence of machine learning in high-stakes applications, especially the ones regulated by anti-discrimination laws or societal norms, it is crucial to ensure that the predictive models do not propagate any existing bias or discrimination. Due to the ability of deep neural nets to learn rich representations, recent advances in algorithmic fairness have focused on learning fair representations with adversarial techniques to reduce bias in data while preserving utility simultaneously. In this paper, through the lens of information theory, we provide the first result that quantitatively characterizes the tradeoff between demographic parity and the joint utility across different population groups. Specifically, when the base rates differ between groups, we show that any method aiming to learn fair representations admits an information-theoretic lower bound on the joint error across these groups. To complement our negative results, we also prove that if the optimal decision functions across different groups are close, then learning fair representations leads to an alternative notion of fairness, known as the accuracy parity, which states that the error rates are close between groups. Finally, our theoretical findings are also confirmed empirically on real-world datasets.

## 1 Introduction

With the prevalence of machine learning applications in high-stakes domains, e.g., criminal judgement, medical testing, online advertising, etc., it is crucial to ensure that the automated decision making systems do not propagate existing bias or discrimination that might exist in historical data [3, 4, 28]. Among many recent proposals for achieving different notions of algorithmic fairness [10, 14, 31–33], learning fair representations has received increasing attention due to recent advances in learning rich representations with deep neural networks [5, 11, 24, 26, 30, 34]. In fact, a line of work has proposed to learn group-invariant representations with adversarial learning techniques in order to achieve statistical parity, also known as the demographic parity in the literature. This line of work dates at least back to Zemel et al. [33] where the authors proposed to learn predictive models that are independent of the group membership attribute. At a high level, the underlying idea is that if representations of instances from different groups are similar to each other, then any predictive model on top of them will certainly make decisions independent of group membership.

On the other hand, it has long been observed that there is an underlying tradeoff between utility and demographic parity:

> *"All methods have in common that to some extent accuracy must be traded-off for lowering the dependency." [6]*

In particular, it is easy to see that in an extreme case where the group membership coincides with the target task, a call for exact demographic parity will inevitably remove the perfect predictor [14].

---

[*]Part of this work was done when Han Zhao was visiting Microsoft Research, Montreal.

Empirically, it has also been observed that a tradeoff exists between accuracy and fairness in binary classification [38]. Clearly, methods based on learning fair representations are also bound by such inherent tradeoff between utility and fairness. But how does the fairness constraint trade for utility? Will learning fair representations help to achieve other notions of fairness besides the demographic parity? If yes, what is the fundamental limit of utility that we can hope to achieve under such constraint?

To answer the above questions, through the lens of information theory, in this paper we provide the first result that quantitatively characterizes the tradeoff between demographic parity and the joint utility across different population groups. Specifically, when the base rates differ between groups, we provide a tight information-theoretic lower bound on the joint error across these groups. Our lower bound is algorithm-independent so it holds for all methods aiming to learn fair representations. When only approximate demographic parity is achieved, we also present a family of lower bounds to quantify the tradeoff of utility introduced by such approximate constraint. As a side contribution, our proof technique is simple but general, and we expect it to have broader applications in other learning problems using adversarial techniques, e.g., unsupervised domain adaptation [12, 36], privacy-preservation under attribute inference attacks [13, 35] and multilingual machine translation [16].

To complement our negative results, we show that if the optimal decision functions across different groups are close, then learning fair representations helps to achieve an alternative notion of fairness, i.e., the accuracy parity, which states that the error rates are close between groups. Empirically, we conduct experiments on a real-world dataset that corroborate both our positive and negative results. We believe our theoretical insights contribute to better understanding of the tradeoff between utility and different notions of fairness, and they are also helpful in guiding the future design of representation learning algorithms to achieve algorithmic fairness.

## 2   Preliminary

We first introduce the notation used throughout the paper and formally describe the problem setup. We then briefly discuss some information-theoretic concepts that will be used in our analysis.

**Notation**    We use $\mathcal{X} \subseteq \mathbb{R}^d$ and $\mathcal{Y} = \{0, 1\}$ to denote the input and output space. Accordingly, we use $X$ and $Y$ to denote the random variables which take values in $\mathcal{X}$ and $\mathcal{Y}$, respectively. Lower case letters $\mathbf{x}$ and $y$ are used to denote the instantiation of $X$ and $Y$. To simplify the presentation, we use $A \in \{0, 1\}$ as the sensitive attribute, e.g., race, gender, etc. [2] Let $\mathcal{H}$ be the hypothesis class of classifiers. In other words, for $h \in \mathcal{H}$, $h : \mathcal{X} \to \mathcal{Y}$ is the predictor that outputs a prediction. Note that even if the predictor does not explicitly take the sensitive attribute $A$ as input, this *fairness through blindness* mechanism can still be biased due to the potential correlations between $X$ and $A$. In this work we study the stochastic setting where there is a joint distribution $\mathcal{D}$ over $X, Y$ and $A$ from which the data are sampled. To keep the notation consistent, for $a \in \{0, 1\}$, we use $\mathcal{D}_a$ to mean the conditional distribution of $\mathcal{D}$ given $A = a$. For an event $E$, $\mathcal{D}(E)$ denotes the probability of $E$ under $\mathcal{D}$. In particular, in the literature of fair machine learning, we call $\mathcal{D}(Y = 1)$ the *base rate* of distribution $\mathcal{D}$ and we use $\Delta_{\mathrm{BR}}(\mathcal{D}, \mathcal{D}') := |\mathcal{D}(Y = 1) - \mathcal{D}'(Y = 1)|$ to denote the difference of the base rates between two distributions $\mathcal{D}$ and $\mathcal{D}'$ over the same sample space. Given a feature transformation function $g : \mathcal{X} \to \mathcal{Z}$ that maps instances from the input space $\mathcal{X}$ to feature space $\mathcal{Z}$, we define $g_\sharp \mathcal{D} := \mathcal{D} \circ g^{-1}$ to be the induced (pushforward) distribution of $\mathcal{D}$ under $g$, i.e., for any event $E' \subseteq \mathcal{Z}$, $g_\sharp \mathcal{D}(E') := \mathcal{D}(g^{-1}(E')) = \mathcal{D}(\{x \in \mathcal{X} \mid g(x) \in E'\})$.

**Problem Setup**    Given a joint distribution $\mathcal{D}$, the error of a predictor $h$ under $\mathcal{D}$ is defined as $\mathrm{Err}_\mathcal{D}(h) := \mathbb{E}_\mathcal{D}[|Y - h(X)|]$. Note that for binary classification problems, when $h(X) \in \{0, 1\}$, $\mathrm{Err}_\mathcal{D}(h)$ reduces to the true error rate of binary classification. To make the notation more compact, we may drop the subscript $\mathcal{D}$ when it is clear from the context. In this work we focus on group fairness where the group membership is given by the sensitive attribute $A$. Even in this context there are many possible definitions of *fairness* [27], and in what follows we provide a brief review of the ones that are mostly relevant to this work.

**Definition 2.1** (Demographic Parity). Given a joint distribution $\mathcal{D}$, a classifier $\widehat{Y}$ satisfies *demographic parity* if $\widehat{Y}$ is independent of $A$.

Demographic parity reduces to the requirement that $\mathcal{D}_0(\widehat{Y} = 1) = \mathcal{D}_1(\widehat{Y} = 1)$, i.e., positive outcome is given to the two groups at the same rate. When exact equality does not hold, we use the absolute difference between them as an approximate measure:

**Definition 2.2** (DP Gap). Given a joint distribution $\mathcal{D}$, the *demographic parity gap* of a classifier $\widehat{Y}$ is $\Delta_{\mathrm{DP}}(\widehat{Y}) := |\mathcal{D}_0(\widehat{Y} = 1) - \mathcal{D}_1(\widehat{Y} = 1)|$.

Demographic parity is also known as *statistical parity*, and it has been adopted as definition of fairness in a series of work [6, 11, 15, 17, 18, 24, 26, 33]. However, as we shall quantify precisely in Section 3, demographic parity may cripple the utility that we hope to achieve, especially in the common scenario where the *base rates* differ between two groups, e.g., $\mathcal{D}_0(Y = 1) \neq \mathcal{D}_1(Y = 1)$ [14]. In light of this, an alternative definition is *accuracy parity*:

**Definition 2.3** (Accuracy Parity). Given a joint distribution $\mathcal{D}$, a classifier $h$ satisfies *accuracy parity* if $\mathrm{Err}_{\mathcal{D}_0}(h) = \mathrm{Err}_{\mathcal{D}_1}(h)$.

In the literature, a break of accuracy parity is also known as disparate mistreatment [32]. Again, when $h$ is a deterministic binary classifier, accuracy parity reduces to $\mathcal{D}_0(h(X) = Y) = \mathcal{D}_1(h(X) = Y)$. Different from demographic parity, the definition of accuracy parity does not eliminate the perfect predictor when $Y = A$ when the base rates differ between two groups. When costs of different error types matter, more refined definitions exist:

**Definition 2.4** (Positive Rate Parity). Given a joint distribution $\mathcal{D}$, a deterministic classifier $h$ satisfies *positive rate parity* if $\mathcal{D}_0(h(X) = 1 \mid Y = y) = \mathcal{D}_1(h(X) = 1 \mid Y = y), \forall y \in \{0, 1\}$.

Positive rate parity is also known as *equalized odds* [14], which essentially requires equal true positive and false positive rates between different groups. Furthermore, Hardt et al. [14] also defined *true positive parity*, or *equal opportunity*, to be $\mathcal{D}_0(h(X) = 1 \mid Y = 1) = \mathcal{D}_1(h(X) = 1 \mid Y = 1)$ when positive outcome is desirable. Last but not least, *predictive rate parity*, also known as *test fairness* [7], asks for equal chance of positive outcomes across groups given predictions:

**Definition 2.5** (Predictive Rate Parity). Given a joint distribution $\mathcal{D}$, a probabilistic classifier $h$ satisfies *predictive rate parity* if $\mathcal{D}_0(Y = 1 \mid h(X) = c) = \mathcal{D}_1(Y = 1 \mid h(X) = c), \forall c \in [0, 1]$.

When $h$ is a deterministic binary classifier that only takes value in $\{0, 1\}$, Chouldechova [7] showed an intrinsic tradeoff between predictive rate parity and positive rate parity:

**Theorem 2.1** (Chouldechova [7]). Assume $\mathcal{D}_0(Y = 1) \neq \mathcal{D}_1(Y = 1)$, then for any deterministic classifier $h : \mathcal{X} \to \{0, 1\}$ that is not perfect, i.e., $h(X) \neq Y$, positive rate parity and predictive rate parity cannot hold simultaneously.

Similar tradeoff result for probabilistic classifier has also been observed by Kleinberg et al. [21], where the authors showed that for any non-perfect predictors, calibration and positive rate parity cannot be achieved simultaneously if the base rates are different across groups. Here a classifier $h$ is said to be *calibrated* if $\mathcal{D}(Y = 1 \mid h(X) = c) = c, \forall c \in [0, 1]$, i.e., if we look at the set of data that receive a predicted probability of $c$ by $h$, we would like $c$-fraction of them to be positive instances according to $Y$ [29].

*f*-**divergence**   Introduced by Ali and Silvey [2] and Csiszár [8, 9], $f$-divergence, also known as the Ali-Silvey distance, is a general class of statistical divergences to measure the difference between two probability distributions $\mathcal{P}$ and $\mathcal{Q}$ over the same measurable space.

**Definition 2.6** ($f$-divergence). Let $\mathcal{P}$ and $\mathcal{Q}$ be two probability distributions over the same space and assume $\mathcal{P}$ is absolutely continuous w.r.t. $\mathcal{Q}$ ($\mathcal{P} \ll \mathcal{Q}$). Then for any convex function $f : (0, \infty) \to \mathbb{R}$ that is strictly convex at 1 and $f(1) = 0$, the $f$-divergence of $\mathcal{Q}$ from $\mathcal{P}$ is defined as

$$D_f(\mathcal{P} \parallel \mathcal{Q}) := \mathbb{E}_{\mathcal{Q}}\left[f\left(\frac{d\mathcal{P}}{d\mathcal{Q}}\right)\right]. \tag{1}$$

The function $f$ is called the *generator function* of $D_f(\cdot \parallel \cdot)$.

Different choices of the generator function $f$ recover popular statistical divergence as special cases, e.g., the KL-divergence. From Jensen's inequality it is easy to verify that $D_f(\mathcal{P} \parallel \mathcal{Q}) \geq 0$ and $D_f(\mathcal{P} \parallel \mathcal{Q}) = 0$ iff $\mathcal{P} = \mathcal{Q}$ almost surely. Note that $f$-divergence does not necessarily leads to

a distance metric, and it is not symmetric in general, i.e., $D_f(\mathcal{P} \| \mathcal{Q}) \neq D_f(\mathcal{Q} \| \mathcal{P})$ provided that $\mathcal{P} \ll \mathcal{Q}$ and $\mathcal{Q} \ll \mathcal{P}$. We list some common choices of the generator function $f$ and their corresponding properties in Table 1. Notably, Khosravifard et al. [20] proved that total variation is the only $f$-divergence that serves as a metric, i.e., satisfying the triangle inequality.

Table 1: List of different $f$-divergences and their corresponding properties. $D_{\mathrm{KL}}(\mathcal{P} \| \mathcal{Q})$ denotes the KL-divergence of $\mathcal{Q}$ from $\mathcal{P}$ and $\mathcal{M} := (\mathcal{P} + \mathcal{Q})/2$ is the average distribution of $\mathcal{P}$ and $\mathcal{Q}$. Symm. stands for Symmetric and Tri. stands for Triangle Inequality.

| Name | $D_f(\mathcal{P} \| \mathcal{Q})$ | Generator $f(t)$ | Symm. | Tri. |
|---|---|---|---|---|
| Kullback-Leibler | $D_{\mathrm{KL}}(\mathcal{P} \| \mathcal{Q})$ | $t \log t$ | ✗ | ✗ |
| Reverse-KL | $D_{\mathrm{KL}}(\mathcal{Q} \| \mathcal{P})$ | $-\log t$ | ✗ | ✗ |
| Jensen-Shannon | $D_{\mathrm{JS}}(\mathcal{P}, \mathcal{Q}) := \frac{1}{2}(D_{\mathrm{KL}}(\mathcal{P}\|\mathcal{M}) + D_{\mathrm{KL}}(\mathcal{Q}\|\mathcal{M}))$ | $t \log t - (t+1)\log(\frac{t+1}{2})$ | ✓ | ✗ |
| Squared Hellinger | $H^2(\mathcal{P}, \mathcal{Q}) := \frac{1}{2}\int(\sqrt{d\mathcal{P}} - \sqrt{d\mathcal{Q}})^2$ | $(1 - \sqrt{t})^2/2$ | ✓ | ✗ |
| Total Variation | $d_{\mathrm{TV}}(\mathcal{P}, \mathcal{Q}) := \sup_E |\mathcal{P}(E) - \mathcal{Q}(E)|$ | $|t - 1|/2$ | ✓ | ✓ |

## 3 Main Results

As we briefly mentioned in Section 2, it is impossible to have imperfect predictor that is both calibrated and preserves positive rate parity when the base rates differ between two groups. Similar impossibility result also holds between positive rate parity and predictive rate parity. On the other hand, while it has long been observed that demographic parity may eliminate perfect predictor [14], and previous work has empirically verified that tradeoff exists between accuracy and demographic parity [6, 17, 38] on various datasets, so far a quantitative characterization on the exact tradeoff between accuracy and various notions of parity is still missing. In what follows we shall prove a family of information theoretic lower bounds on the accuracy that hold for *all* algorithms.

### 3.1 Tradeoff between Fairness and Utility

Essentially, every prediction function induces a Markov chain: $X \xrightarrow{g} Z \xrightarrow{h} \widehat{Y}$, where $g$ is the feature transformation, $h$ is the classifier on feature space, $Z$ is the feature and $\widehat{Y}$ is the predicted target variable by $h \circ g$. Note that simple models, e.g., linear classifiers, are also included by specifying $g$ to be the identity map. With this notation, we first state the following theorem that quantifies an inherent tradeoff between fairness and utility.

**Theorem 3.1.** Let $\widehat{Y} = h(g(X))$ be the predictor. If $\widehat{Y}$ satisfies demographic parity, then $\mathrm{Err}_{\mathcal{D}_0}(h \circ g) + \mathrm{Err}_{\mathcal{D}_1}(h \circ g) \geq \Delta_{\mathrm{BR}}(\mathcal{D}_0, \mathcal{D}_1)$.

**Remark**   First of all, $\Delta_{\mathrm{BR}}(\mathcal{D}_0, \mathcal{D}_1)$ is the difference of base rates across groups, and it achieves its maximum value of 1 iff $Y = A$ almost surely, i.e., $Y$ indicates group membership. On the other hand, if $Y$ is independent of $A$, then $\Delta_{\mathrm{BR}}(\mathcal{D}_0, \mathcal{D}_1) = 0$ so the lower bound does not make any constraint on the joint error. Second, Theorem 3.1 applies to all possible feature transformation $g$ and predictor $h$. In particular, if we choose $g$ to be the identity map, then Theorem 3.1 says that when the base rates differ, *no algorithm* can achieve a small joint error on both groups, and it also recovers the previous observation that demographic parity can eliminate the perfect predictor [14]. Third, the lower bound in Theorem 3.1 is insensitive to the marginal distribution of $A$, i.e., it treats the errors from both groups equally. As a comparison, let $\alpha := \mathcal{D}(A = 1)$, then $\mathrm{Err}_{\mathcal{D}}(h \circ g) = (1 - \alpha)\mathrm{Err}_{\mathcal{D}_0}(h \circ g) + \alpha\mathrm{Err}_{\mathcal{D}_1}(h \circ g)$. In this case $\mathrm{Err}_{\mathcal{D}}(h \circ g)$ could still be small even if the minority group suffers a large error.

Furthermore, by the pigeonhole principle, the following corollary holds:

**Corollary 3.1.** If the predictor $\widehat{Y} = h(g(X))$ satisfies demographic parity, then $\max\{\mathrm{Err}_{\mathcal{D}_0}(h \circ g), \mathrm{Err}_{\mathcal{D}_1}(h \circ g)\} \geq \Delta_{\mathrm{BR}}(\mathcal{D}_0, \mathcal{D}_1)/2$.

In words, this means that for fair predictors in the demographic parity sense, at least one of the subgroups has to incur an error of at least $\Delta_{\mathrm{BR}}(\mathcal{D}_0, \mathcal{D}_1)/2$ which could be large in settings like criminal justice where $\Delta_{\mathrm{BR}}(\mathcal{D}_0, \mathcal{D}_1)$ is large.

Before we give the proof, we first present a useful lemma that lower bounds the prediction error by the total variation distance.

**Lemma 3.1.** Let $\widehat{Y} = h(g(X))$ be the predictor, then for $a \in \{0, 1\}$, $d_{\mathrm{TV}}(\mathcal{D}_a(Y), \mathcal{D}_a(\widehat{Y})) \leq \mathrm{Err}_{\mathcal{D}_a}(h \circ g)$.

*Proof.* For $a \in \{0, 1\}$, we have:

$$
\begin{aligned}
d_{\mathrm{TV}}(\mathcal{D}_a(Y), \mathcal{D}_a(\widehat{Y})) &= |\mathcal{D}_a(Y = 1) - \mathcal{D}_a(h(g(X)) = 1)| = |\mathbb{E}_{\mathcal{D}_a}[Y] - \mathbb{E}_{\mathcal{D}_a}[h(g(X))]| \\
&\leq \mathbb{E}_{\mathcal{D}_a}[|Y - h(g(X))|] = \mathrm{Err}_{\mathcal{D}_a}(h \circ g).
\end{aligned}
$$

∎

Now we are ready to prove Theorem 3.1:

*Proof of Theorem 3.1.* First of all, we show that if $\widehat{Y} = h(g(X))$ satisfies demographic parity, then:

$$
\begin{aligned}
d_{\mathrm{TV}}(\mathcal{D}_0(\widehat{Y}), \mathcal{D}_1(\widehat{Y})) &= \max\left\{|\mathcal{D}_0(\widehat{Y} = 0) - \mathcal{D}_1(\widehat{Y} = 0)|,\ |\mathcal{D}_0(\widehat{Y} = 1) - \mathcal{D}_1(\widehat{Y} = 1)|\right\} \\
&= |\mathcal{D}_0(\widehat{Y} = 1) - \mathcal{D}_1(\widehat{Y} = 1)| \\
&= |\mathcal{D}(\widehat{Y} = 1 \mid A = 0) - \mathcal{D}(\widehat{Y} = 1 \mid A = 1)| = 0,
\end{aligned}
$$

where the last equality follows from the definition of demographic parity. Now from Table 1, $d_{\mathrm{TV}}(\cdot, \cdot)$ is symmetric and satisfies the triangle inequality, we have:

$$
\begin{aligned}
d_{\mathrm{TV}}(\mathcal{D}_0(Y), \mathcal{D}_1(Y)) &\leq d_{\mathrm{TV}}(\mathcal{D}_0(Y), \mathcal{D}_0(\widehat{Y})) + d_{\mathrm{TV}}(\mathcal{D}_0(\widehat{Y}), \mathcal{D}_1(\widehat{Y})) + d_{\mathrm{TV}}(\mathcal{D}_1(\widehat{Y}), \mathcal{D}_1(Y)) \\
&= d_{\mathrm{TV}}(\mathcal{D}_0(Y), \mathcal{D}_0(\widehat{Y})) + d_{\mathrm{TV}}(\mathcal{D}_1(\widehat{Y}), \mathcal{D}_1(Y)). \quad (2)
\end{aligned}
$$

The last step is to bound $d_{\mathrm{TV}}(\mathcal{D}_a(Y), \mathcal{D}_a(\widehat{Y}))$ in terms of $\mathrm{Err}_{\mathcal{D}_a}(h \circ g)$ for $a \in \{0, 1\}$ using Lemma 3.1:

$$
d_{\mathrm{TV}}(\mathcal{D}_0(Y), \mathcal{D}_0(\widehat{Y})) \leq \mathrm{Err}_{\mathcal{D}_0}(h \circ g), \quad d_{\mathrm{TV}}(\mathcal{D}_1(Y), \mathcal{D}_1(\widehat{Y})) \leq \mathrm{Err}_{\mathcal{D}_1}(h \circ g).
$$

Combining the above two inequalities and (2) completes the proof. ∎

It is not hard to show that our lower bound in Theorem 3.1 is tight. To see this, consider the case $A = Y$, where the lower bound achieves its maximum value of 1. Now consider a constant predictor $\widehat{Y} \equiv 1$ or $\widehat{Y} \equiv 0$, which clearly satisfies demographic parity by definition. But in this case either $\mathrm{Err}_{\mathcal{D}_0}(h \circ g) = 1, \mathrm{Err}_{\mathcal{D}_1}(h \circ g) = 0$ or $\mathrm{Err}_{\mathcal{D}_0}(h \circ g) = 0, \mathrm{Err}_{\mathcal{D}_1}(h \circ g) = 1$, hence $\mathrm{Err}_{\mathcal{D}_0}(h \circ g) + \mathrm{Err}_{\mathcal{D}_1}(h \circ g) \equiv 1$, achieving the lower bound.

To conclude this section, we point out that the choice of total variation in the lower bound is not unique. As we will see shortly in Section 3.2, similar lower bounds could be attained using specific choices of the general $f$-divergence with some desired properties.

## 3.2 Tradeoff in Fair Representation Learning

In the last section we show that there is an inherent tradeoff between fairness and utility when a predictor *exactly* satisfies demographic parity. In practice we may not be able to achieve demographic parity exactly. Instead, most algorithms [1, 5, 11, 24] build an adversarial discriminator that takes as input the feature vector $Z = g(X)$, and the goal is to learn fair representations such that it is hard for the adversarial discriminator to infer the group membership from $Z$. In this sense due to the limit on the capacity of the adversarial discriminator, only approximate demographic parity can be achieved in the equilibrium. Hence it is natural to ask what is the tradeoff between fair representations and accuracy in this scenario? In this section we shall answer this question by generalizing our previous analysis with $f$-divergence to prove a family of lower bounds on the joint target prediction error. Our results also show how approximate DP helps to reconcile but not remove the tradeoff between fairness and utility. Before we state and prove the main results in this section, we first introduce the following lemma by Liese and Vajda [22] as a generalization of the data processing inequality for $f$-divergence:

**Lemma 3.2** (Liese and Vajda [22])**.** Let $\mu(\mathcal{Z})$ be the space of all probability distributions over $\mathcal{Z}$. Then for any $f$-divergence $D_f(\cdot \| \cdot)$, any stochastic kernel $\kappa : \mathcal{X} \to \mu(\mathcal{Z})$, and any distributions $\mathcal{P}$ and $\mathcal{Q}$ over $\mathcal{X}$, $D_f(\kappa \mathcal{P} \| \kappa \mathcal{Q}) \leq D_f(\mathcal{P} \| \mathcal{Q})$.

Roughly speaking, Lemma 3.2 says that data processing cannot increase discriminating information. Define $d_{\mathrm{JS}}(\mathcal{P}, \mathcal{Q}) := \sqrt{D_{\mathrm{JS}}(\mathcal{P}, \mathcal{Q})}$ and $H(\mathcal{P}, \mathcal{Q}) := \sqrt{H^2(\mathcal{P}, \mathcal{Q})}$. It is well-known in information theory that both $d_{\mathrm{JS}}(\cdot, \cdot)$ and $H(\cdot, \cdot)$ form a bounded distance metric over the space of probability distributions. Realize that $d_{\mathrm{TV}}(\cdot, \cdot)$, $H^2(\cdot, \cdot)$ and $D_{\mathrm{JS}}(\cdot, \cdot)$ are all $f$-divergence. The following corollary holds:

**Corollary 3.2.** Let $h : \mathcal{Z} \to \mathcal{Y}$ be any hypothesis, and $g_\sharp \mathcal{D}_a$ be the pushforward distribution of $\mathcal{D}_a$ by $g$, $\forall a \in \{0, 1\}$. Let $\widehat{Y} = h(g(X))$ be the predictor, then all the following inequalities hold:

1. $d_{\mathrm{TV}}(\mathcal{D}_0(\widehat{Y}), \mathcal{D}_1(\widehat{Y})) \leq d_{\mathrm{TV}}(g_\sharp \mathcal{D}_0, g_\sharp \mathcal{D}_1)$
2. $H(\mathcal{D}_0(\widehat{Y}), \mathcal{D}_1(\widehat{Y})) \leq H(g_\sharp \mathcal{D}_0, g_\sharp \mathcal{D}_1)$
3. $d_{\mathrm{JS}}(\mathcal{D}_0(\widehat{Y}), \mathcal{D}_1(\widehat{Y})) \leq d_{\mathrm{JS}}(g_\sharp \mathcal{D}_0, g_\sharp \mathcal{D}_1)$

Now we are ready to present the following main theorem of this section:

**Theorem 3.2.** Let $\widehat{Y} = h(g(X))$ be the predictor. Assume $d_{\mathrm{JS}}(g_\sharp \mathcal{D}_0, g_\sharp \mathcal{D}_1) \leq d_{\mathrm{JS}}(\mathcal{D}_0(Y), \mathcal{D}_1(Y))$ and $H(g_\sharp \mathcal{D}_0, g_\sharp \mathcal{D}_1) \leq H(\mathcal{D}_0(Y), \mathcal{D}_1(Y))$, then the following three inequalities hold:

1. Total variation lower bound:
$$\mathrm{Err}_{\mathcal{D}_0}(h \circ g) + \mathrm{Err}_{\mathcal{D}_1}(h \circ g) \geq d_{\mathrm{TV}}(\mathcal{D}_0(Y), \mathcal{D}_1(Y)) - d_{\mathrm{TV}}(g_\sharp \mathcal{D}_0, g_\sharp \mathcal{D}_1).$$

2. Jensen-Shannon lower bound:
$$\mathrm{Err}_{\mathcal{D}_0}(h \circ g) + \mathrm{Err}_{\mathcal{D}_1}(h \circ g) \geq \left(d_{\mathrm{JS}}(\mathcal{D}_0(Y), \mathcal{D}_1(Y)) - d_{\mathrm{JS}}(g_\sharp \mathcal{D}_0, g_\sharp \mathcal{D}_1)\right)^2 / 2.$$

3. Hellinger lower bound:
$$\mathrm{Err}_{\mathcal{D}_0}(h \circ g) + \mathrm{Err}_{\mathcal{D}_1}(h \circ g) \geq \left(H(\mathcal{D}_0(Y), \mathcal{D}_1(Y)) - H(g_\sharp \mathcal{D}_0, g_\sharp \mathcal{D}_1)\right)^2 / 2.$$

**Remark**   All the three lower bounds in Theorem 3.2 imply a tradeoff between the joint error across demographic subgroups and learning group-invariant feature representations. In a nutshell:

> *For fair representations, it is not possible to construct a predictor that simultaneously minimizes the errors on both demographic subgroups.*

When $g_\sharp \mathcal{D}_0 = g_\sharp \mathcal{D}_1$, which also implies $\mathcal{D}_0(\widehat{Y}) = \mathcal{D}_1(\widehat{Y})$, all three lower bounds get larger, in this case we have

$$\max \left\{ d_{\mathrm{TV}}(\mathcal{D}_0(Y), \mathcal{D}_1(Y)), \frac{1}{2} d_{\mathrm{JS}}^2(\mathcal{D}_0(Y), \mathcal{D}_1(Y)), \frac{1}{2} H^2(\mathcal{D}_0(Y), \mathcal{D}_1(Y)) \right\} = d_{\mathrm{TV}}(\mathcal{D}_0(Y), \mathcal{D}_1(Y))$$
$$= \Delta_{\mathrm{BR}}(\mathcal{D}_0, \mathcal{D}_1),$$

and this reduces to Theorem 3.1. Now we give a sketch of the proof for Theorem 3.2:

*Proof Sketch of Theorem 3.2.* We prove the three inequalities respectively. The total variation lower bound follows the same idea as the proof of Theorem 3.1 and the inequality $d_{\mathrm{TV}}(\mathcal{D}_0(\widehat{Y}), \mathcal{D}_1(\widehat{Y})) \leq d_{\mathrm{TV}}(g_\sharp \mathcal{D}_0, g_\sharp \mathcal{D}_1)$ from Corollary 3.2. To prove the Jensen-Shannon lower bound, realize that $d_{\mathrm{JS}}(\cdot, \cdot)$ is a distance metric over probability distributions. Combining with the inequality $d_{\mathrm{JS}}(\mathcal{D}_0(\widehat{Y}), \mathcal{D}_1(\widehat{Y})) \leq d_{\mathrm{JS}}(g_\sharp \mathcal{D}_0, g_\sharp \mathcal{D}_1)$ from Corollary 3.2, we have:

$$d_{\mathrm{JS}}(\mathcal{D}_0(Y), \mathcal{D}_1(Y)) \leq d_{\mathrm{JS}}(\mathcal{D}_0(Y), \mathcal{D}_0(\widehat{Y})) + d_{\mathrm{JS}}(g_\sharp \mathcal{D}_0, g_\sharp \mathcal{D}_1) + d_{\mathrm{JS}}(\mathcal{D}_1(\widehat{Y}), \mathcal{D}_1(Y)).$$

Now by Lin's lemma [23, Theorem 3], for any two distributions $\mathcal{P}$ and $\mathcal{Q}$, we have $d_{\mathrm{JS}}^2(\mathcal{P}, \mathcal{Q}) \leq d_{\mathrm{TV}}(\mathcal{P}, \mathcal{Q})$. Combine Lin's lemma with Lemma 3.1, we get the following lower bound:

$$\sqrt{\mathrm{Err}_{\mathcal{D}_0}(h \circ g)} + \sqrt{\mathrm{Err}_{\mathcal{D}_1}(h \circ g)} \geq d_{\mathrm{JS}}(\mathcal{D}_0(Y), \mathcal{D}_1(Y)) - d_{\mathrm{JS}}(g_\sharp \mathcal{D}_0, g_\sharp \mathcal{D}_1).$$

Apply the AM-GM inequality, we can further bound the L.H.S. by

$$\sqrt{2\left(\mathrm{Err}_{\mathcal{D}_0}(h \circ g) + \mathrm{Err}_{\mathcal{D}_1}(h \circ g)\right)} \geq \sqrt{\mathrm{Err}_{\mathcal{D}_0}(h \circ g)} + \sqrt{\mathrm{Err}_{\mathcal{D}_1}(h \circ g)}.$$

Under the assumption that $d_{\mathrm{JS}}(g_\sharp \mathcal{D}_0, g_\sharp \mathcal{D}_1) \leq d_{\mathrm{JS}}(\mathcal{D}_0(Y), \mathcal{D}_1(Y))$, taking a square at both sides then completes the proof for the second inequality. The proof for Hellinger's lower bound follows exactly as the one for Jensen-Shannon's lower bound, except that instead of Lin's lemma, we need to use the fact that $H^2(\mathcal{P}, \mathcal{Q}) \leq d_{\mathrm{TV}}(\mathcal{P}, \mathcal{Q}) \leq \sqrt{2} H(\mathcal{P}, \mathcal{Q})$, $\forall \mathcal{P}, \mathcal{Q}$. ∎

As a simple corollary of Theorem 3.2, the following result shows how approximate DP (in terms of the DP gap) helps to reconcile the tradeoff between fairness and utility:

**Corollary 3.3.** Let $\widehat{Y} = h(g(X))$ be the predictor, then $\mathrm{Err}_{\mathcal{D}_0}(h \circ g) + \mathrm{Err}_{\mathcal{D}_1}(h \circ g) \geq \Delta_{\mathrm{BR}}(\mathcal{D}_0, \mathcal{D}_1) - \Delta_{\mathrm{DP}}(\widehat{Y})$.

In a sense Corollary 3.3 means that in order to lower the joint error, the DP gap of the predictor cannot be too small. Of course, since the above inequality is a lower bound, it only serves as a necessary condition for small joint error. Hence an interesting question would be to ask whether it is possible to have a sufficient condition that guarantees a small joint error such that the DP gap of the predictor is no larger than that of the perfect predictor, i.e., $\Delta_{\mathrm{BR}}(\mathcal{D}_0, \mathcal{D}_1)$. We leave this as a future work.

## 3.3 Fair Representations Lead to Accuracy Parity

In the previous sections we prove a family of information-theoretic lower bounds that demonstrate an inherent tradeoff between fair representations and joint error across groups. A natural question to ask then, is, what kind of parity can fair representations bring us? To complement our negative results, in this section we show that learning group-invariant representations help to reduce discrepancy of errors (utilities) across groups.

First of all, since we work under the stochastic setting where $\mathcal{D}_a$ is a joint distribution over $X$ and $Y$ conditioned on $A = a$, then any function mapping $h : \mathcal{X} \to \mathcal{Y}$ will inevitably incur an error due to the noise existed in the distribution $\mathcal{D}_a$. Formally, for $a \in \{0, 1\}$, define the optimal function $h_a^* : \mathcal{X} \to \mathcal{Y}$ under the absolute error to be $h_a^*(X) := m_{\mathcal{D}_a}(Y \mid X)$, where $m_{\mathcal{D}_a}(Y \mid X)$ denotes the median of $Y$ given $X$ under distribution $\mathcal{D}_a$. Now define the noise of distribution $\mathcal{D}_a$ to be $n_{\mathcal{D}_a} := \mathbb{E}_{\mathcal{D}_a}[|Y - h_a^*(X)|]$. With these notations, we are now ready to present the following theorem:

**Theorem 3.3.** For any hypothesis $\mathcal{H} \ni h : \mathcal{X} \to \mathcal{Y}$, the following inequality holds:

$$|\mathrm{Err}_{\mathcal{D}_0}(h) - \mathrm{Err}_{\mathcal{D}_1}(h)| \leq n_{\mathcal{D}_0} + n_{\mathcal{D}_1} + d_{\mathrm{TV}}(\mathcal{D}_0(X), \mathcal{D}_1(X))$$
$$+ \min \left\{ \mathbb{E}_{\mathcal{D}_0}[|h_0^* - h_1^*|], \mathbb{E}_{\mathcal{D}_1}[|h_0^* - h_1^*|] \right\}.$$

**Remark**  Theorem 3.3 upper bounds the discrepancy of accuracy across groups by three terms: the noise, the distance of representations across groups and the discrepancy of optimal decision functions. In an ideal setting where both distributions are noiseless, i.e., same people in the same group are always treated equally, the upper bound simplifies to the latter two terms:

$$|\mathrm{Err}_{\mathcal{D}_0}(h) - \mathrm{Err}_{\mathcal{D}_1}(h)| \leq d_{\mathrm{TV}}(\mathcal{D}_0(X), \mathcal{D}_1(X)) + \min \left\{ \mathbb{E}_{\mathcal{D}_0}[|h_0^* - h_1^*|], \mathbb{E}_{\mathcal{D}_1}[|h_0^* - h_1^*|] \right\}.$$

If we further require that the optimal decision functions $h_0^*$ and $h_1^*$ are close to each other, i.e., optimal decisions are insensitive to the group membership, then Theorem 3.3 implies that a sufficient condition to guarantee accuracy parity is to find group-invariant representation that minimizes $d_{\mathrm{TV}}(\mathcal{D}_0(X), \mathcal{D}_1(X))$. We now present the proof for Theorem 3.3:

*Proof of Theorem 3.3.* First, we show that for $a \in \{0, 1\}$, $\mathrm{Err}_{\mathcal{D}_a}(h)$ cannot be too large if $h$ is close to $h_a^*$:

$$|\mathrm{Err}_{\mathcal{D}_a}(h) - n_{\mathcal{D}_a}| = |\mathrm{Err}_{\mathcal{D}_a}(h) - \mathrm{Err}_{\mathcal{D}_a}(h_a^*)| = \left| \mathbb{E}_{\mathcal{D}_a}[|Y - h(X)|] - \mathbb{E}_{\mathcal{D}_a}[|Y - h_a^*(X)|] \right|$$
$$\leq \mathbb{E}_{\mathcal{D}_a}[|h(X) - h_a^*(X)|],$$

where the inequality is due to triangular inequality. Next, we bound $|\mathrm{Err}_{\mathcal{D}_0}(h) - \mathrm{Err}_{\mathcal{D}_1}(h)|$ by:

$$|\mathrm{Err}_{\mathcal{D}_0}(h) - \mathrm{Err}_{\mathcal{D}_1}(h)| \leq n_{\mathcal{D}_0} + n_{\mathcal{D}_1} + \left| \mathbb{E}_{\mathcal{D}_0}[|h(X) - h_0^*(X)|] - \mathbb{E}_{\mathcal{D}_1}[|h(X) - h_1^*(X)|] \right|.$$

In order to show this, define $\varepsilon_a(h, h') := \mathbb{E}_{\mathcal{D}_a}[|h(X) - h'(X)|]$ so that

$$\left| \mathbb{E}_{\mathcal{D}_0}[|h(X) - h_0^*(X)|] - \mathbb{E}_{\mathcal{D}_1}[|h(X) - h_1^*(X)|] \right| = \left| \varepsilon_0(h, h_0^*) - \varepsilon_1(h, h_1^*) \right|.$$

To bound $\left| \varepsilon_0(h, h_0^*) - \varepsilon_1(h, h_1^*) \right|$, realize that $|h(X) - h_a^*(X)| \in \{0, 1\}$. On one hand, we have:

$$\left| \varepsilon_0(h, h_0^*) - \varepsilon_1(h, h_1^*) \right| = \left| \varepsilon_0(h, h_0^*) - \varepsilon_0(h, h_1^*) + \varepsilon_0(h, h_1^*) - \varepsilon_1(h, h_1^*) \right|$$
$$\leq \left| \varepsilon_0(h, h_0^*) - \varepsilon_0(h, h_1^*) \right| + \left| \varepsilon_0(h, h_1^*) - \varepsilon_1(h, h_1^*) \right|$$
$$\leq \varepsilon_0(h_0^*, h_1^*) + d_{\mathrm{TV}}(\mathcal{D}_0(X), \mathcal{D}_1(X)),$$

where the last inequality is due to $\left|\varepsilon_0(h, h_1^*) - \varepsilon_1(h, h_1^*)\right| = \left|\mathcal{D}_0(|h - h_1^*| = 1) - \mathcal{D}_1(|h - h_1^*| = 1)\right| \leq \sup_E |\mathcal{D}_0(E) - \mathcal{D}_1(E)| = d_{\text{TV}}(\mathcal{D}_0, \mathcal{D}_1)$. Similarly, by subtracting and adding back $\varepsilon_1(h, h_0^*)$ instead, we can also show that $\left|\varepsilon_0(h, h_0^*) - \varepsilon_1(h, h_1^*)\right| \leq \varepsilon_1(h_0^*, h_1^*) + d_{\text{TV}}(\mathcal{D}_0(X), \mathcal{D}_1(X))$.

Combine the above two inequalities yielding:

$$\left|\varepsilon_0(h, h_0^*) - \varepsilon_1(h, h_1^*)\right| \leq \min\{\varepsilon_0(h_0^*, h_1^*), \varepsilon_1(h_0^*, h_1^*)\} + d_{\text{TV}}(\mathcal{D}_0(X), \mathcal{D}_1(X)).$$

Incorporating the two noise terms back to the above inequality then completes the proof. ∎

## 4  Empirical Validation

Our theoretical results on the lower bound imply that over-training the feature transformation function to achieve group-invariant representations will inevitably lead to large joint errors. On the other hand, our upper bound also implies that group-invariant representations help to achieve accuracy parity. To verify these theoretical implications, in this section we conduct experiments on a real-world benchmark dataset, the UCI Adult dataset, to present empirical results with various metrics.

**Dataset**  The Adult dataset contains 30,162/15,060 training/test instances for income prediction. Each instance in the dataset describes an adult from the 1994 US Census. Attributes include gender, education level, age, etc. In this experiment we use gender (binary) as the sensitive attribute, and we preprocess the dataset to convert categorical variables into one-hot representations. The processed data contains 114 attributes. The target variable (income) is also binary: 1 if $\geq$ 50K/year otherwise 0. For the sensitive attribute $A$, $A = 0$ means Male otherwise Female. In this dataset, the base rates across groups are different: $\Pr(Y = 1 \mid A = 0) = 0.310$ while $\Pr(Y = 1 \mid A = 1) = 0.113$. Also, the group ratios are different: $\Pr(A = 0) = 0.673$.

**Experimental Protocol**  To validate the effect of learning group-invariant representations with adversarial debiasing techniques [5, 26, 34], we perform a controlled experiment by fixing the baseline network architecture to be a three hidden-layer feed-forward network with ReLU activations. The number of units in each hidden layer are 500, 200, and 100, respectively. The output layer corresponds to a logistic regression model. This baseline without debiasing is denoted as NoDebias. For debiasing with adversarial learning techniques, the adversarial discriminator network takes the feature from the last hidden layer as input, and connects it to a hidden-layer with 50 units, followed by a binary classifier whose goal is to predict the sensitive attribute $A$. This model is denoted as AdvDebias. Compared with NoDebias, the only difference of AdvDebias in terms of objective function is that besides the cross-entropy loss for target prediction, the AdvDebias also contains a classification loss from the adversarial discriminator to predict the sensitive attribute $A$. In the experiment, all the other factors are fixed to be the same between these two methods, including learning rate, optimization algorithm, training epoch, and also batch size. To see how the adversarial loss affects the joint error, the demographic parity as well as the accuracy parity, we vary the coefficient $\rho$ for the adversarial loss between 0.1, 1.0, 5.0 and 50.0.

**Results and Analysis**  The experimental results are listed in Table 2. Note that in the table $|\text{Err}_{\mathcal{D}_0} - \text{Err}_{\mathcal{D}_1}|$ could be understood as measuring an approximate version of accuracy parity, and similarly $\Delta_{\text{DP}}(\widehat{Y})$ measures the closeness of the classifier to satisfy demographic parity. From the table, it is then clear that with increasing $\rho$, both the overall error $\text{Err}_{\mathcal{D}}$ (sensitive to the marginal distribution of $A$) and the joint error $\text{Err}_{\mathcal{D}_0} + \text{Err}_{\mathcal{D}_1}$ (insensitive to the imbalance of $A$) are increasing. As expected, $\Delta_{\text{DP}}(\widehat{Y})$ is drastically decreasing with the increasing of $\rho$. Furthermore, $|\text{Err}_{\mathcal{D}_0} - \text{Err}_{\mathcal{D}_1}|$ is also gradually decreasing, but much slowly than $\Delta_{\text{DP}}(\widehat{Y})$. This is due to the existing noise in the data as well as the shift between the optimal decision functions across groups, as indicated by our upper bound. To conclude, all the empirical results are consistent with our theoretical findings.

## 5  Related Work

**Fairness Frameworks**  Two central notions of fairness have been extensively studied, i.e., group fairness and individual fairness. In a seminal work, Dwork et al. [10] define individual fairness as a measure of smoothness of the classification function. Under the assumption that number of

Table 2: Adversarial debiasing on demographic parity, joint error across groups, and accuracy parity.

|  | $\text{Err}_{\mathcal{D}}$ | $\text{Err}_{\mathcal{D}_0} + \text{Err}_{\mathcal{D}_1}$ | $|\text{Err}_{\mathcal{D}_0} - \text{Err}_{\mathcal{D}_1}|$ | $\Delta_{\text{DP}}(\widehat{Y})$ |
|---|---|---|---|---|
| NoDebias | 0.157 | 0.275 | 0.115 | 0.189 |
| AdvDebias, $\rho = 0.1$ | 0.159 | 0.278 | 0.116 | 0.190 |
| AdvDebias, $\rho = 1.0$ | 0.162 | 0.286 | 0.106 | 0.113 |
| AdvDebias, $\rho = 5.0$ | 0.166 | 0.295 | 0.106 | 0.032 |
| AdvDebias, $\rho = 50.0$ | 0.201 | 0.360 | 0.112 | 0.028 |

individuals is finite, the authors proposed a linear programming framework to maximize the utility under their fairness constraint. However, their framework requires apriori a distance function that computes the similarity between individuals, and their optimization formulation does not produce an inductive rule to generalize to unseen data. Based on the definition of positive rate parity, Hardt et al. [14] proposed a post-processing method to achieve fairness by taking as input the prediction and the sensitive attribute. In a concurrent work, Kleinberg et al. [21] offer a calibration technique to achieve the corresponding fairness criterion as well. However, both of the aforementioned two approaches require sensitive attribute during the inference phase, which is not available in many real-world scenarios.

**Regularization Techniques** The line of work on fairness-aware learning through regularization dates at least back to Kamishima et al. [19], where the authors argue that simple deletion of sensitive features in data is insufficient for eliminating biases in automated decision making, due to the possible correlations among attributes and sensitive information [25]. In light of this, the authors proposed a *prejudice remover* regularizer that essentially penalizes the mutual information between the predicted goal and the sensitive information. In a more recent approach, Zafar et al. [31] leveraged a measure of decision boundary fairness and incorporated it via constraints into the objective function of logistic regression as well as support vector machines. As discussed in Section 2, both approaches essentially reduce to achieving demographic parity through regularization.

**Representation Learning** In a pioneer work, Zemel et al. [33] proposed to preserve both group and individual fairness through the lens of representation learning, where the main idea is to find a good representation of the data with two competing goals: to encode the data for utility maximization while at the same time to obfuscate any information about membership in the protected group. Due to the power of learning rich representations offered by deep neural nets, recent advances in building fair automated decision making systems focus on using adversarial techniques to learn fair representation that also preserves enough information for the prediction vendor to achieve his utility [1, 5, 11, 24, 30, 34, 37]. Madras et al. [26] further extended this approach by incorporating reconstruction loss given by an autoencoder into the objective function to preserve demographic parity, equalized odds, and equal opportunity.

## 6 Conclusion

In this paper we theoretically and empirically study the important problem of quantifying the tradeoff between utility and fairness in learning group-invariant representations. Specifically, we prove a novel lower bound to characterize the tradeoff between demographic parity and the joint utility across different population groups when the base rates differ between groups. In particular, our results imply that any method aiming to learn fair representations admits an information-theoretic lower bound on the joint error, and the better the representation, the larger the joint error. Complementary to our negative results, we also show that learning fair representations leads to accuracy parity if the optimal decision functions across different groups are close. These theoretical findings are also confirmed empirically on real-world datasets. We believe our results take an important step towards better understanding the tradeoff between utility and different notions of fairness. Inspired by our lower bound, one interesting direction for future work is to design instance-weighting algorithm to balance the base rates during representation learning.

**Acknowledgments**

HZ and GG would like to acknowledge support from the DARPA XAI project, contract #FA87501720152 and an Nvidia GPU grant. HZ would also like to thank Jianfeng Chi for helpful discussions on the relationship between algorithmic fairness and privacy-preservation learning.

## Footnotes

[2]Our main results could also be straightforwardly extended to the setting where $A$ is a categorical variable.

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
