[Supplementary Material]

# Inherent Tradeoffs in Learning Fair Representation

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

* the methods. Due to space limit, we defer most of the proofs to appendix, while only leaving one to showcase the high-level idea of our proof technique.

## 3.1 Tradeoff between Accuracy and Demographic Parity

Essentially, every prediction function induces a Markov chain: $X \xrightarrow{g} Z \xrightarrow{h} \hat{Y}$, where $g$ is the feature transformation, $h$ is the classifier on feature space, $Z$ is the feature and $\hat{Y}$ is the predicted target variable by $h \circ g$. Note that simple models, e.g., linear classifiers, are also included by specifying $g$ to be the identity map. With this notation, we first state the following theorem that quantifies an information-theoretic lower bound on the joint error across different groups:

**Theorem 3.1.** Let $\hat{Y} = h(g(X))$ be the predictor. If $\hat{Y}$ satisfies demographic parity, then $\mathrm{Err}_{\mathcal{D}_0}(h \circ g) + \mathrm{Err}_{\mathcal{D}_1}(h \circ g) \geq d_{\mathrm{TV}}(\mathcal{D}_0(Y), \mathcal{D}_1(Y))$.

**Remark** First of all, $d_{\mathrm{TV}}(\mathcal{D}_0(Y), \mathcal{D}_1(Y))$ essentially measures the discrepancy of base rates across groups, and $d_{\mathrm{TV}}(\mathcal{D}_0(Y), \mathcal{D}_1(Y))$ achieves its maximum value of 1 iff $Y = A$ almost surely, i.e., $Y$ indicates group membership. Second, Theorem 3.1 applies to all possible feature transformation $g$ and predictor $h$. In particular, if we choose $g$ to be the identity map, then Theorem 3.1 says that when the base rates differ, *no algorithm* can achieve a small joint error on both groups, and it also recovers the previous observation that demographic parity can eliminate the perfect predictor [17]. Third, the lower bound in Theorem 3.1 is insensitive to the marginal distribution of $A$, i.e., it treats the errors from both groups equally. As a comparison, let $\alpha := \mathcal{D}(A = 1)$, then $\mathrm{Err}_{\mathcal{D}}(h \circ g) = (1 - \alpha)\mathrm{Err}_{\mathcal{D}_0}(h \circ g) + \alpha\mathrm{Err}_{\mathcal{D}_1}(h \circ g)$. In this case $\mathrm{Err}_{\mathcal{D}}(h \circ g)$ could still be small even if the minority group suffers a large error.

Before we give the proof, we first present a useful lemma that lower bounds the prediction error by the total variation distance.

**Lemma 3.1.** Let $\hat{Y} = h(g(X))$ be the predictor, then for $a \in \{0, 1\}$, $d_{\mathrm{TV}}(\mathcal{D}_a(Y), \mathcal{D}_a(\hat{Y})) \leq \mathrm{Err}_{\mathcal{D}_a}(h \circ g)$.

*Proof of Theorem 3.1.* First of all, we show that if $\hat{Y} = h(g(X))$ satisfies demographic parity, then:

$$d_{\mathrm{TV}}(\mathcal{D}_0(\hat{Y}), \mathcal{D}_1(\hat{Y})) = \max\left\{\mid\mathcal{D}_0(\hat{Y} = 0) - \mathcal{D}_1(\hat{Y} = 0)\mid, \mid\mathcal{D}_0(\hat{Y} = 1) - \mathcal{D}_1(\hat{Y} = 1)\mid\right\}$$
$$= \mid\mathcal{D}_0(\hat{Y} = 1) - \mathcal{D}_1(\hat{Y} = 1)\mid$$
$$= \mid\mathcal{D}(\hat{Y} = 1 \mid A = 0) - \mathcal{D}(\hat{Y} = 1 \mid A = 1)\mid = 0,$$

where the last equality follows from the definition of demographic parity. Now from Table 1, $d_{\mathrm{TV}}(\cdot, \cdot)$ is symmetric and satisfies the triangle inequality, we have:

$$d_{\mathrm{TV}}(\mathcal{D}_0(Y), \mathcal{D}_1(Y)) \leq d_{\mathrm{TV}}(\mathcal{D}_0(Y), \mathcal{D}_0(\hat{Y})) + d_{\mathrm{TV}}(\mathcal{D}_0(\hat{Y}), \mathcal{D}_1(\hat{Y})) + d_{\mathrm{TV}}(\mathcal{D}_1(\hat{Y}), \mathcal{D}_1(Y))$$
$$= d_{\mathrm{TV}}(\mathcal{D}_0(Y), \mathcal{D}_0(\hat{Y})) + d_{\mathrm{TV}}(\mathcal{D}_1(\hat{Y}), \mathcal{D}_1(Y)). \tag{2}$$

The last step is to bound $d_{\mathrm{TV}}(\mathcal{D}_a(Y), \mathcal{D}_a(\hat{Y}))$ in terms of $\mathrm{Err}_{\mathcal{D}_a}(h \circ g)$ for $a \in \{0, 1\}$ using Lemma 3.1:

$$d_{\mathrm{TV}}(\mathcal{D}_0(Y), \mathcal{D}_0(\hat{Y})) \leq \mathrm{Err}_{\mathcal{D}_0}(h \circ g), \quad d_{\mathrm{TV}}(\mathcal{D}_1(Y), \mathcal{D}_1(\hat{Y})) \leq \mathrm{Err}_{\mathcal{D}_1}(h \circ g).$$

Combining the above two inequalities and (2) completes the proof. ∎

It is not hard to show that our lower bound in Theorem 3.1 is tight. To see this, consider the case $A = Y$, where the lower bound achieves its maximum value $d_{\mathrm{TV}}(\mathcal{D}_0(Y), \mathcal{D}_1(Y)) = 1$. Now consider a constant predictor $\hat{Y} \equiv 1$ or $\hat{Y} \equiv 0$, which clearly satisfies demographic parity by definition. But in this case either $\mathrm{Err}_{\mathcal{D}_0}(h \circ g) = 1, \mathrm{Err}_{\mathcal{D}_1}(h \circ g) = 0$ or $\mathrm{Err}_{\mathcal{D}_0}(h \circ g) = 0, \mathrm{Err}_{\mathcal{D}_1}(h \circ g) = 1$, hence $\mathrm{Err}_{\mathcal{D}_0}(h \circ g) + \mathrm{Err}_{\mathcal{D}_1}(h \circ g) \equiv 1$, achieving the lower bound.

To conclude this subsection, we point out that the choice of total variation in the lower bound is not unique. As we will see shortly, similar lower bounds could be attained using specific choices of the general $f$-divergence with some desired properties.

## 3.2 Tradeoff in Adversarial Representation Learning

Theorem 3.1 is an impossibility result on achieving accuracy and exact demographic parity jointly. But what if we only aim to achieve approximate demographic parity? What is the tradeoff between demographic parity and accuracy in this scenario? In fact, from the perspective of representation learning, recent work [6, 14, 29, 38] have proposed to learn intermediate feature $Z$ through deep neural networks, aiming to maintain task-relevant information while at the same time removing sensitive information related to $A$. To study the tradeoff in this setting, we introduce a relaxed version of total variation, known as the $\mathcal{H}$-divergence [4]:

**Definition 3.1** ($\mathcal{H}$-divergence). Let $\mathcal{H}$ be a hypothesis class on feature space $\mathcal{Z}$, and $\mathcal{A}_{\mathcal{H}}$ be the collection of subsets of $\mathcal{Z}$ that are the support of some hypothesis in $\mathcal{H}$, i.e., $\mathcal{A}_{\mathcal{H}} := \{h^{-1}(1) \mid h \in \mathcal{H}\}$. The distance between two distributions $\mathcal{D}$ and $\mathcal{D}'$ over $\mathcal{Z}$ based on $\mathcal{H}$ is: $d_{\mathcal{H}}(\mathcal{D}, \mathcal{D}') := \sup_{A \in \mathcal{A}_{\mathcal{H}}} |\mathcal{D}(A) - \mathcal{D}'(A)|$.

$\mathcal{H}$-divergence is particularly favorable in the analysis of adversarial representation learning with binary classification problems, and it had also been generalized to the *discrepancy distance* [10, 30] for general loss functions. When $\mathcal{H}$ has a finite VC-dimension, $\mathcal{H}$-divergence can be estimated using finite unlabeled samples from $\mathcal{D}$ and $\mathcal{D}'$ [23]. From an algorithmic viewpoint, $\mathcal{H}$-divergence admits a natural interpretation that $1 - d_{\mathcal{H}}(\mathcal{D}, \mathcal{D}')$ corresponds to the minimum sum of Type-I and Type-II error in distinguishing $\mathcal{D}$ and $\mathcal{D}'$. To see this, realize

$$
\begin{aligned}
1 - d_{\mathcal{H}}(\mathcal{D}, \mathcal{D}') &= 1 - \sup_{A \in \mathcal{A}_{\mathcal{H}}} |\mathcal{D}(A) - \mathcal{D}'(A)| \\
&= \inf_{A \in \mathcal{A}_{\mathcal{H}}} 1 - \mathcal{D}(A) + \mathcal{D}'(A) = \inf_{h \in \mathcal{H}} \mathcal{D}(h(Z) = 0) + \mathcal{D}'(h(Z) = 1). \quad (3)
\end{aligned}
$$

The second equality holds because for $h \in \mathcal{H}$, we also have $1 - h \in \mathcal{H}$. In (3) the hypothesis $h$ acts as a discriminator trying to distinguish between $\mathcal{D}$ and $\mathcal{D}'$. The above probabilistic interpretation exactly serves as the theoretical justification of recent work on using adversarial training to learn group-invariant representation $Z$ through transformation $g$ such that $d_{\mathcal{H}}(\mathcal{D}_0^g, \mathcal{D}_1^g)$ is small, where $\mathcal{D}_a^g$ is the induced distribution of $\mathcal{D}_a$ under $g$. The following proposition exactly characterizes an intrinsic tradeoff of these methods:

**Proposition 3.1.** Let $\hat{Y} = h(g(X))$ be the predictor and $\lambda_{\mathcal{H}} := 1 - d_{\mathcal{H}}(\mathcal{D}_0(\hat{Y}), \mathcal{D}_1(\hat{Y}))$. Then $\mathrm{Err}_{\mathcal{D}_0}(h \circ g) + \mathrm{Err}_{\mathcal{D}_1}(h \circ g) \geq d_{\mathcal{H}}(\mathcal{D}_0(Y), \mathcal{D}_1(Y)) + \lambda_{\mathcal{H}} - 1$.

**Remark** As we show above, $\lambda_{\mathcal{H}}$ is the minimum sum of Type-I and Type-II errors in discriminating $\mathcal{D}_0(\hat{Y})$ and $\mathcal{D}_1(\hat{Y})$ using discriminators from $\mathcal{H}$. Hence if the optimal discriminator from $\mathcal{H}$ fails to distinguish between $\mathcal{D}_0(\hat{Y})$ and $\mathcal{D}_1(\hat{Y})$, i.e., larger $\lambda_{\mathcal{H}}$, the lower bound on the joint error across different groups will also get larger.

In fact, a close scrutiny of the proof above shows that the lower bound in Proposition 3.1 holds even if different transformation functions are used on the corresponding groups:

**Corollary 3.1.** Let $\hat{Y} = h(g_a(X))$ be the predictors for group $A = a, a \in \{0, 1\}$ and $\lambda_{\mathcal{H}}$ as defined in Proposition 3.1. Then $\mathrm{Err}_{\mathcal{D}_0}(h \circ g_0) + \mathrm{Err}_{\mathcal{D}_1}(h \circ g_1) \geq d_{\mathcal{H}}(\mathcal{D}_0(Y), \mathcal{D}_1(Y)) + \lambda_{\mathcal{H}} - 1$.

One interesting fact implied by Proposition 3.1 is that the lower bound of the joint error across groups scales linearly with $\lambda_{\mathcal{H}}$, the optimal sum of Type-I and Type-II errors in distinguishing between $\mathcal{D}_0(\hat{Y})$ and $\mathcal{D}_1(\hat{Y})$. In the work of Zhang et al. [38], the authors proposed a model (Fig. 1 in [38]) that precisely tries to maximize $\lambda_{\mathcal{H}}$ by learning the model parameters of $g$ through adversarial techniques. In this case our lower bound directly quantifies the loss of utility due to the increase of $\lambda_{\mathcal{H}}$.

## 3.3 A Family of Information-Theoretic Lower Bounds

In the last subsection we show that any adversarial discriminator that tries to distinguish between $\mathcal{D}_0(\hat{Y})$ and $\mathcal{D}_1(\hat{Y})$ by taking the predicted target variable $\hat{Y}$ as input admits an inherent lower bound in terms of joint target error. This is the algorithm proposed by Zhang et al. [38] for mitigating biases. As a comparison, most other variants [1, 6, 14, 27] build an adversarial discriminator that takes as input the feature vector $Z = g(X)$ instead. In this subsection we generalize our previous analysis with $f$-divergence to prove a family of lower bounds on the joint target prediction error for the latter variants. Based on our theoretical analysis, we conclude that matching the distributions from different groups within the feature space does not remove the tradeoff. In fact, a family of lower bounds also exist for these approaches.

We require one last piece of ingredient before we state and prove the main results in this section. The following lemma is proved by Liese and Vajda [25] as a generalization of the data processing inequality for $f$-divergence:

**Lemma 3.2** (Liese and Vajda [25]). Let $\mu(\mathcal{Z})$ be the space of all probability distributions over $\mathcal{Z}$. Then for any $f$-divergence $D_f(\cdot \,||\, \cdot)$, any stochastic kernel $\kappa : \mathcal{X} \to \mu(\mathcal{Z})$, and any distributions $\mathcal{P}$ and $\mathcal{Q}$ over $\mathcal{X}$, $D_f(\kappa\mathcal{P} \,||\, \kappa\mathcal{Q}) \leq D_f(\mathcal{P} \,||\, \mathcal{Q})$.

Roughly speaking, Lemma 3.2 says that data processing cannot increase discriminating information. Define $d_{\mathrm{JS}}(\mathcal{P}, \mathcal{Q}) := \sqrt{D_{\mathrm{JS}}(\mathcal{P}, \mathcal{Q})}$ and $H(\mathcal{P}, \mathcal{Q}) := \sqrt{H^2(\mathcal{P}, \mathcal{Q})}$. Both $d_{\mathrm{JS}}(\cdot, \cdot)$ and $H(\cdot, \cdot)$ form a bounded distance metric over the space of probability distributions. Realize that $d_{\mathrm{TV}}(\cdot, \cdot)$, $H^2(\cdot, \cdot)$ and $D_{\mathrm{JS}}(\cdot, \cdot)$ are all $f$-divergence. The following corollary holds:

**Corollary 3.2.** Let $h : \mathcal{Z} \to \mathcal{Y}$ to any (randomized) hypothesis, and $\mathcal{D}_a^g$ be the induced distribution of $\mathcal{D}_a$ by $g$, $\forall a \in \{0, 1\}$. Let $\hat{Y} = h(g(X))$ be the predictor, then 1). $d_{\mathrm{TV}}(\mathcal{D}_0(\hat{Y}), \mathcal{D}_1(\hat{Y})) \leq d_{\mathrm{TV}}(\mathcal{D}_0^g, \mathcal{D}_1^g)$. 2). $H(\mathcal{D}_0(\hat{Y}), \mathcal{D}_1(\hat{Y})) \leq H(\mathcal{D}_0^g, \mathcal{D}_1^g)$. 3). $d_{\mathrm{JS}}(\mathcal{D}_0(\hat{Y}), \mathcal{D}_1(\hat{Y})) \leq d_{\mathrm{JS}}(\mathcal{D}_0^g, \mathcal{D}_1^g)$.

Now we are ready to present the following main theorem of this subsection:

**Theorem 3.2.** Let $\hat{Y} = h(g(X))$ be the predictor. Assume $d_{\mathrm{JS}}(\mathcal{D}_0^g, \mathcal{D}_1^g) \leq d_{\mathrm{JS}}(\mathcal{D}_0(Y), \mathcal{D}_1(Y))$ and $H(\mathcal{D}_0^g, \mathcal{D}_1^g) \leq H(\mathcal{D}_0(Y), \mathcal{D}_1(Y))$, then the following three inequalities hold:

1. Total variation lower bound:

$$\mathrm{Err}_{\mathcal{D}_0}(h \circ g) + \mathrm{Err}_{\mathcal{D}_1}(h \circ g) \geq d_{\mathrm{TV}}(\mathcal{D}_0(Y), \mathcal{D}_1(Y)) - d_{\mathrm{TV}}(\mathcal{D}_0^g, \mathcal{D}_1^g).$$

2. Jensen-Shannon lower bound:

$$\mathrm{Err}_{\mathcal{D}_0}(h \circ g) + \mathrm{Err}_{\mathcal{D}_1}(h \circ g) \geq \left(d_{\mathrm{JS}}(\mathcal{D}_0(Y), \mathcal{D}_1(Y)) - d_{\mathrm{JS}}(\mathcal{D}_0^g, \mathcal{D}_1^g)\right)^2 / 2.$$

3. Hellinger lower bound:

$$\mathrm{Err}_{\mathcal{D}_0}(h \circ g) + \mathrm{Err}_{\mathcal{D}_1}(h \circ g) \geq \left(H(\mathcal{D}_0(Y), \mathcal{D}_1(Y)) - H(\mathcal{D}_0^g, \mathcal{D}_1^g)\right)^2 / 2.$$

**Remark** All the three lower bounds in Theorem 3.2 imply a tradeoff between demographic parity and the joint error across groups through learning group-invariant feature representations. When $\mathcal{D}_0^g = \mathcal{D}_1^g$, which also implies $\mathcal{D}_0(\hat{Y}) = \mathcal{D}_1(\hat{Y})$, all three lower bounds get larger, in this case we have $\max\{d_{\mathrm{TV}}(\mathcal{D}_0(Y), \mathcal{D}_1(Y)), \frac{1}{2}d_{\mathrm{JS}}^2(\mathcal{D}_0(Y), \mathcal{D}_1(Y)), \frac{1}{2}H^2(\mathcal{D}_0(Y), \mathcal{D}_1(Y))\} = d_{\mathrm{TV}}(\mathcal{D}_0(Y), \mathcal{D}_1(Y))$, and this reduces to the tight lower bound in Theorem 3.1.

## 3.4 Group-Invariant Representation Leads to Accuracy Parity

In previous subsections we prove a family of information-theoretic lower bounds that demonstrate an inherent tradeoff between demographic parity and joint error across groups. Specifically, we show that group-invariant representation will also inevitably compromise utility. A natural question to ask then, is, what kind of parity can group-invariant representation bring us? To complement our negative results, in this subsection we show that learning group-invariant representation helps to reduce discrepancy of errors (utilities) across groups.

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

The experimental results are listed in Table 2. Note that in the table $|\mathrm{Err}_{\mathcal{D}_0} - \mathrm{Err}_{\mathcal{D}_1}|$ could be understood as measuring an approximate version of accuracy parity, and similarly $d_{\mathrm{TV}}(\mathcal{D}_0(\hat{Y}), \mathcal{D}_1(\hat{Y}))$ measures the closeness of the prediction function to demographic parity. From the table, it is then clear that with increasing $\rho$, both the overall error $\mathrm{Err}_{\mathcal{D}}$ (sensitive to the marginal distribution of $A$) and the joint error $\mathrm{Err}_{\mathcal{D}_0} + \mathrm{Err}_{\mathcal{D}_1}$ (insensitive to the imbalance of $A$) are increasing. As expected, $d_{\mathrm{TV}}(\mathcal{D}_0(\hat{Y}), \mathcal{D}_1(\hat{Y}))$ is drastically decreasing with the increasing of $\rho$. Furthermore, $|\mathrm{Err}_{\mathcal{D}_0} - \mathrm{Err}_{\mathcal{D}_1}|$ is also gradually decreasing, but much slowly than $d_{\mathrm{TV}}(\mathcal{D}_0(\hat{Y}), \mathcal{D}_1(\hat{Y}))$. This is due to the existing noise in the data as well as the shift between the optimal decision functions across groups, as indicated by our upper bound in Theorem 3.3. To conclude, all the empirical results are consistent with our theoretical findings.

Table 2: Adversarial debiasing on demographic parity, joint error across groups, and accuracy parity.

| | $\text{Err}_{\mathcal{D}}$ | $\text{Err}_{\mathcal{D}_0} + \text{Err}_{\mathcal{D}_1}$ | $|\text{Err}_{\mathcal{D}_0} - \text{Err}_{\mathcal{D}_1}|$ | $d_{\text{TV}}(\mathcal{D}_0(\hat{Y}), \mathcal{D}_1(\hat{Y}))$ |
|---|---|---|---|---|
| NoDebias | 0.157 | 0.275 | 0.115 | 0.189 |
| AdvDebias, $\rho = 0.1$ | 0.159 | 0.278 | 0.116 | 0.190 |
| AdvDebias, $\rho = 1.0$ | 0.162 | 0.286 | 0.106 | 0.113 |
| AdvDebias, $\rho = 5.0$ | 0.166 | 0.295 | 0.106 | 0.032 |

## 5  Related Work

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

∎

**Proposition 3.1.** Let $\hat{Y} = h(g(X))$ be the predictor and $\lambda_{\mathcal{H}} := 1 - d_{\mathcal{H}}(\mathcal{D}_0(\hat{Y}), \mathcal{D}_1(\hat{Y}))$. Then $\mathrm{Err}_{\mathcal{D}_0}(h \circ g) + \mathrm{Err}_{\mathcal{D}_1}(h \circ g) \geq d_{\mathcal{H}}(\mathcal{D}_0(Y), \mathcal{D}_1(Y)) + \lambda_{\mathcal{H}} - 1$.

*Proof.* First, it is easy to show that $\forall \mathcal{H}$ and any distributions $\mathcal{D}, \mathcal{D}'$ and $\mathcal{D}''$, the following triangle inequality holds:

$$
\begin{aligned}
d_{\mathcal{H}}(\mathcal{D}, \mathcal{D}') = \sup_{A \in \mathcal{A}_{\mathcal{H}}} |\mathcal{D}(A) - \mathcal{D}'(A)| &\leq \sup_{A \in \mathcal{A}_{\mathcal{H}}} |\mathcal{D}(A) - \mathcal{D}''(A)| + |\mathcal{D}''(A) - \mathcal{D}'(A)| \\
&\leq \sup_{A \in \mathcal{A}_{\mathcal{H}}} |\mathcal{D}(A) - \mathcal{D}''(A)| + \sup_{A \in \mathcal{A}_{\mathcal{H}}} |\mathcal{D}''(A) - \mathcal{D}'(A)| = d_{\mathcal{H}}(\mathcal{D}, \mathcal{D}'') + d_{\mathcal{H}}(\mathcal{D}'', \mathcal{D}').
\end{aligned}
$$

Again, we apply a chain of triangle inequalities as we did in the proof of Theorem 3.1:

$$
\begin{aligned}
d_{\mathcal{H}}(\mathcal{D}_0(Y), \mathcal{D}_1(Y)) &\leq d_{\mathcal{H}}(\mathcal{D}_0(Y), \mathcal{D}_0(\hat{Y})) + d_{\mathcal{H}}(\mathcal{D}_0(\hat{Y}), \mathcal{D}_1(\hat{Y})) + d_{\mathcal{H}}(\mathcal{D}_1(\hat{Y}), \mathcal{D}_1(Y)) \\
&\leq d_{\mathrm{TV}}(\mathcal{D}_0(Y), \mathcal{D}_0(\hat{Y})) + d_{\mathrm{TV}}(\mathcal{D}_1(\hat{Y}), \mathcal{D}_1(Y)) + d_{\mathcal{H}}(\mathcal{D}_1(\hat{Y}), \mathcal{D}_1(Y)) \\
&\leq \mathrm{Err}_{\mathcal{D}_0}(h \circ g) + \mathrm{Err}_{\mathcal{D}_1}(h \circ g) + 1 - \lambda_{\mathcal{H}}.
\end{aligned}
$$

The second inequality follows from the definition of $\mathcal{H}$-divergence and the last one is due to (4). Arranging the terms on both sides completes the proof. ∎

**Theorem 3.2.** Let $\hat{Y} = h(g(X))$ be the predictor. Assume $d_{\mathrm{JS}}(\mathcal{D}_0^g, \mathcal{D}_1^g) \leq d_{\mathrm{JS}}(\mathcal{D}_0(Y), \mathcal{D}_1(Y))$ and $H(\mathcal{D}_0^g, \mathcal{D}_1^g) \leq H(\mathcal{D}_0(Y), \mathcal{D}_1(Y))$, then the following three inequalities hold:

1. Total variation lower bound:
$$
\mathrm{Err}_{\mathcal{D}_0}(h \circ g) + \mathrm{Err}_{\mathcal{D}_1}(h \circ g) \geq d_{\mathrm{TV}}(\mathcal{D}_0(Y), \mathcal{D}_1(Y)) - d_{\mathrm{TV}}(\mathcal{D}_0^g, \mathcal{D}_1^g).
$$

2. Jensen-Shannon lower bound:
$$
\mathrm{Err}_{\mathcal{D}_0}(h \circ g) + \mathrm{Err}_{\mathcal{D}_1}(h \circ g) \geq \big(d_{\mathrm{JS}}(\mathcal{D}_0(Y), \mathcal{D}_1(Y)) - d_{\mathrm{JS}}(\mathcal{D}_0^g, \mathcal{D}_1^g)\big)^2/2.
$$

3. Hellinger lower bound:
$$
\mathrm{Err}_{\mathcal{D}_0}(h \circ g) + \mathrm{Err}_{\mathcal{D}_1}(h \circ g) \geq \big(H(\mathcal{D}_0(Y), \mathcal{D}_1(Y)) - H(\mathcal{D}_0^g, \mathcal{D}_1^g)\big)^2/2.
$$

*Proof.* We prove the three inequalities respectively. The total variation lower bound follows the same idea as the proof of Theorem 3.1 and the inequality $d_{\mathrm{TV}}(\mathcal{D}_0(\hat{Y}), \mathcal{D}_1(\hat{Y})) \leq d_{\mathrm{TV}}(\mathcal{D}_0^g, \mathcal{D}_1^g)$ from Corollary 3.2. To prove the Jensen-Shannon lower bound, realize that $d_{\mathrm{JS}}(\cdot, \cdot)$ is a distance metric over probability distributions. Combining with the inequality $d_{\mathrm{JS}}(\mathcal{D}_0(\hat{Y}), \mathcal{D}_1(\hat{Y})) \leq d_{\mathrm{JS}}(\mathcal{D}_0^g, \mathcal{D}_1^g)$ from Corollary 3.2, we have:

$$
d_{\mathrm{JS}}(\mathcal{D}_0(Y), \mathcal{D}_1(Y)) \leq d_{\mathrm{JS}}(\mathcal{D}_0(Y), \mathcal{D}_0(\hat{Y})) + d_{\mathrm{JS}}(\mathcal{D}_0^g, \mathcal{D}_1^g) + d_{\mathrm{JS}}(\mathcal{D}_1(\hat{Y}), \mathcal{D}_1(Y)).
$$

Now by Lin's lemma [26, Theorem 3], for any two distributions $\mathcal{P}$ and $\mathcal{Q}$, we have $d_{\mathrm{JS}}^2(\mathcal{P}, \mathcal{Q}) \leq d_{\mathrm{TV}}(\mathcal{P}, \mathcal{Q})$. Combine Lin's lemma with Lemma 3.1, we get the following lower bound:

$$
\sqrt{\mathrm{Err}_{\mathcal{D}_0}(h \circ g)} + \sqrt{\mathrm{Err}_{\mathcal{D}_1}(h \circ g)} \geq d_{\mathrm{JS}}(\mathcal{D}_0(Y), \mathcal{D}_1(Y)) - d_{\mathrm{JS}}(\mathcal{D}_0^g, \mathcal{D}_1^g).
$$

Apply a simple AM-GM inequality, we can further bound the L.H.S. by

$$
\sqrt{2\big(\mathrm{Err}_{\mathcal{D}_0}(h \circ g) + \mathrm{Err}_{\mathcal{D}_1}(h \circ g)\big)} \geq \sqrt{\mathrm{Err}_{\mathcal{D}_0}(h \circ g)} + \sqrt{\mathrm{Err}_{\mathcal{D}_1}(h \circ g)}.
$$

Under the assumption that $d_{\mathrm{JS}}(\mathcal{D}_0^g, \mathcal{D}_1^g) \leq d_{\mathrm{JS}}(\mathcal{D}_0(Y), \mathcal{D}_1(Y))$, taking the square at both sides then completes the proof for the second inequality. The proof for Hellinger's lower bound follows exactly as the one for Jensen-Shannon's lower bound, except that we need to use $H^2(\mathcal{P}, \mathcal{Q}) \leq d_{\mathrm{TV}}(\mathcal{P}, \mathcal{Q}) \leq \sqrt{2}H(\mathcal{P}, \mathcal{Q})$, $\forall \mathcal{P}, \mathcal{Q}$, instead of Lin's lemma. ∎

488 **Theorem 3.3.** For any hypothesis $\mathcal{H} \ni h : \mathcal{X} \to \mathcal{Y}$, the following inequality holds:

$$|\text{Err}_{\mathcal{D}_0}(h) - \text{Err}_{\mathcal{D}_1}(h)| \leq n_{\mathcal{D}_0} + n_{\mathcal{D}_1} + d_{\text{TV}}(\mathcal{D}_0(X), \mathcal{D}_1(X)) + \min\{\mathbb{E}_{\mathcal{D}_0}[|h_0^* - h_1^*|], \mathbb{E}_{\mathcal{D}_1}[|h_0^* - h_1^*|]\}.$$

489
490 *Proof.* First, we show that for $a \in \{0, 1\}$, $\text{Err}_{\mathcal{D}_a}(h)$ cannot be too large if $h$ is close to $h_a^*$:

$$|\text{Err}_{\mathcal{D}_a}(h) - n_{\mathcal{D}_a}| = |\text{Err}_{\mathcal{D}_a}(h) - \text{Err}_{\mathcal{D}_a}(h_a^*)| = \left|\mathbb{E}_{\mathcal{D}_a}[|Y - h(X)|] - \mathbb{E}_{\mathcal{D}_a}[|Y - h_a^*(X)|]\right|$$
$$\leq \mathbb{E}_{\mathcal{D}_a}[|h(X) - h_a^*(X)|].$$

491 Next, we bound $|\text{Err}_{\mathcal{D}_0}(h) - \text{Err}_{\mathcal{D}_1}(h)|$ by:

$$|\text{Err}_{\mathcal{D}_0}(h) - \text{Err}_{\mathcal{D}_1}(h)| \leq n_{\mathcal{D}_0} + n_{\mathcal{D}_1} + \left|\mathbb{E}_{\mathcal{D}_0}[|h(X) - h_0^*(X)|] - \mathbb{E}_{\mathcal{D}_1}[|h(X) - h_1^*(X)|]\right|.$$

492 To simplify the notation, define $\varepsilon_a(h, h') := \mathbb{E}_{\mathcal{D}_a}[|h(X) - h'(X)|]$ so that

$$\left|\mathbb{E}_{\mathcal{D}_0}[|h(X) - h_0^*(X)|] - \mathbb{E}_{\mathcal{D}_1}[|h(X) - h_1^*(X)|]\right| = \left|\varepsilon_0(h, h_0^*) - \varepsilon_1(h, h_1^*)\right|.$$

493 To bound $\left|\varepsilon_0(h, h_0^*) - \varepsilon_1(h, h_1^*)\right|$, realize that $|h(X) - h_a^*(X)| \in \{0, 1\}$. On one hand, we have:

$$\left|\varepsilon_0(h, h_0^*) - \varepsilon_1(h, h_1^*)\right| = \left|\varepsilon_0(h, h_0^*) - \varepsilon_0(h, h_1^*) + \varepsilon_0(h, h_1^*) - \varepsilon_1(h, h_1^*)\right|$$
$$\leq \left|\varepsilon_0(h, h_0^*) - \varepsilon_0(h, h_1^*)\right| + \left|\varepsilon_0(h, h_1^*) - \varepsilon_1(h, h_1^*)\right|$$
$$\leq \varepsilon_0(h_0^*, h_1^*) + d_{\text{TV}}(\mathcal{D}_0(X), \mathcal{D}_1(X)),$$

494 where the last inequality is due to $\left|\varepsilon_0(h, h_1^*) - \varepsilon_1(h, h_1^*)\right| = \left|\mathcal{D}_0(|h - h_1^*| = 1) - \mathcal{D}_1(|h - h_1^*| = 1)\right| \leq$
495 $\sup_E |\mathcal{D}_0(E) - \mathcal{D}_1(E)| = d_{\text{TV}}(\mathcal{D}_0, \mathcal{D}_1)$. Similarly, by subtracting and adding back $\varepsilon_1(h, h_0^*)$ instead,
496 we can also show that $\left|\varepsilon_0(h, h_0^*) - \varepsilon_1(h, h_1^*)\right| \leq \varepsilon_1(h_0^*, h_1^*) + d_{\text{TV}}(\mathcal{D}_0(X), \mathcal{D}_1(X))$. Combining all
497 the inequalities above finishes the proof. ∎