[Reviews · NeurIPS 2019]

Reviewer 1



EDIT: I've read the rebuttal. My opinion/score of the paper hasn't changed: I still think the results are interesting, but the authors spend too long discussing the simpler results, and not long enough discussing the more complex/interesting results. However, the rebuttal has convinced me that the "simpler" results are more novel than I originally understood. Thanks for giving the extra line for Table 2. It demonstrates a continuation of the pattern for Error and D_TV, but not for accuracy parity, which seems unusual, so I'm not sure what to make of it. In this work, the authors present several information-theoretic bounds relating accuracy and fairness in the fair representation learning setup. They discuss properties of binary predictors, relating their accuracy, fairness (under demographic parity), and underlying base rates. They relate some of these properties to learned representations. Finally they present bounds relating the learned representation to the induced error learned by a post-hoc classifier on a downstream task. They also present a result regarding the differing levels of accuracy achieved on two groups from a learned representation. Finally, they present a small experimental section to compare to their theoretical results. The central results in the paper are in Sections 3.3 and 3.4. These results deal most directly with the nature of the learned representations – they are both the most interesting and the most novel results. The first result gives lower bounds on the error (weighted inversely by group size) in terms of several f-divergences between the learned representations for the two groups. This is a nice result. In particular, I haven’t seen the Liese and Vadja lemma used in this setting before; that seems useful. One question I have is regarding dimensionality of the representation space – when are these divergences feasible to calculate? For instance, in a high-dimensional continuous space, TVD is intractable. It would be interesting for the authors to address this question with some discussion. The main result of Section 3.4 is regarding accuracy parity – the gap in error rates achieved on the two groups in the data. This is, in my opinion, an understudied fairness metric, and I was pleased to see some work on it here. Roughly, the accuracy parity is upper bounded in terms of the inherent noise of the data + the distance between learned representations + the similarity of the two groups in the data. One thing is not clear to me: the TV term in this bound seems to be distance in X space (D_0(x), D_1(X)), but the authors describe it as distance in Z space – which is true? The earlier theoretical work (Sec. 3.1 and 3.2) regarding the properties of binary classifiers is, in my opinion, less interesting and less novel. As the authors state on line 219, these sections consider an adversary which only looks at Yhat, rather than the representation. This is a much less rich setup. These results seem very related to results regarding lower bounds on demographic parity from the cited papers by Madras et al. and Zhang et al. – if not overlapping, I would like to see some discussion on the relationship. It’s not so clear to me that these results really have much to do with the paper as presented – they are not really about representation learning. In this sense, it’s a shame that Sec 3.1 and 3.2 are much longer than the more interesting and relevant 3.3 and 3.4. Ideally, I would like to see more exploration of the ideas in 3.3 and 3.4 instead. Notes: - Title: should be plural – “Inherent Tradeoffs in Learning Fair Representations” - Line 44: I don’t’ think this is the first result quantitatively characterizing the fairness-accuracy tradeoff. See Menon and Williamson “The Cost of Fairness in Binary Classification”. I’d recommend making a more precise claim, or clarifying. - Line 128: I may be unfamiliar with notation here: what does it mean that P << Q and also Q << P? - Line 134: typo: “A similar impossibility…” - Table 1: should the Generator for KL divergence be logt, not tlogt? - Line 273: When can we expect optimal decisions to be insensitive to group membership? How does this differ from having identical distributions to begin with? - Experiments: This paper is mainly theoretical, which is fine. I’m not sure the experiments are very enlightening. It might be helpful to try more values of lambda and show a more consistent trend, or try to work in a setting where you can verify the bounds tractably.

Reviewer 2



The paper "Inherent Tradeoffs in Learning Fair Representation" analyses the trade-off between learning fair representations and the resulting loss for the targetted task. The analysis is done via lowerbounds of the prediction error from fair representations, by the use of f-divergence measures. It also gives an upper-bound of the discrepancy of accuracy across groups. My feeling is that the paper is interesting, theoretically well sounded and rather well written, but the findings are too much obvious. It has a great pedagogical interest, but do we really need all of this theoretical analysis to state that when gaining in fairness tend to reduce the prediction ability ? To strengthen the contribution, it would be required to propose a novel approach based on the bounds derived in the paper. However it is one of the first papers that gives a solid theoretical analysis of trade-offs between fairness and utility. While findings are not surprising, bounds given in the paper can inspire researchers in fairness. Authors of the paper give a possible direction in conclusion. Based on the other reviews and the authors feedback, I changed my mark from 4 to 6.

Reviewer 3



After Author Rebuttal ================ I agree with acceptance. It's nice to see such a clearly-written paper! That said I would love for the authors to include a discussion in the final version of the paper that addresses some of these questions in my review: I'd like to understand better how this relates to the results of Chouldechova [9] and Kleinberg et al., [24]. - Is the result here stronger? - Would we expect solutions, such as the suggested instance-weighting future work, to circumvent the above results as well? I'd also like to know how these results generalize - Do the results here extend to equal opportunity? - Would collecting additional data circumvent this result? And more about future work: - Do you think the lower bound is a useful tool to understand base distribution drift over time? - Why does instance-weighting improve fairness? ================ 1. Summary This paper gives a lower bound on the error of fair representations when base rates are different, if they aren't, the fair representations lead to accuracy parity. 2. High level paper This paper is very clearly written. I think there are sufficient experiments. This paper nicely complements the trade-off results of Chouldechova [9] and Kleinberg et al., [24]. It will certainly be interesting to people. 3. High level technical I enjoyed reading this paper. I have no complaints, but there are a few things I'd like to understand better: I'd like to understand better how this relates to the results of Chouldechova [9] and Kleinberg et al., [24]. - Is the result here stronger? - Would we expect solutions, such as the suggested instance-weighting future work, to circumvent the above results as well? I'd also like to know how these results generalize - Do the results here extend to equal opportunity? - Would collecting additional data circumvent this result? And more about future work: - Do you think the lower bound is a useful tool to understand base distribution drift over time? - Why does instance-weighting improve fairness? 4. Review summary Overall, this paper is well-suited for publication at neurips.

[Author Response · NeurIPS 2019]

We thank all the reviewers for the time devoted to provide thoughtful comments.

**[Reviewer # 1, Estimating information-theoretic distances]** First, we would like to thank the reviewer for the comprehensive and accurate summary of our work. We are happy that the reviewer found our results to be novel and useful. We agree with the reviewer that estimating the distance between the learned representations are intractable, since the sample complexity of estimating Shannon entropy, mutual information and related concepts are exponential in the dimension of the representation space. That being said, one alternative way to do so is to consider the variational representations of $f$-divergence and use rich parametrized function class (e.g., neural networks) to approximate these distances. For example, recent work [2] on estimating mutual information has empirically shown that such approach often leads to better estimation result than classic approaches based on nonparametric density estimation. From this perspective, the $\mathcal{H}$-divergence in Section 3.2 actually serves as a relaxation of the total variation distance and it equals TVD when $\mathcal{H}$ contains all the measurable functions. Hence the lower bound in Proposition 3.1 gives us a practical way to estimate a proxy of TVD in terms of sum of Type-I and Type-II errors in distinguishing group memberships.

**[Reviewer # 1, Total variation in Theorem 3.3]** The total variation in Theorem 3.3 is w.r.t the input distributions across groups, i.e., $\mathcal{D}_0(X)$ and $\mathcal{D}_1(X)$. In the remark we use "the distance of representation" to mean "the distance of input distributions". We will clarify this sentence in our final version to avoid such confusion.

**[Reviewer # 1, Comparisons with existing work]** The results in this paper are distinct from the results in [1]. Specifically, the trade-off given in [1] (Proposition 8) is in terms of the fairness frontier function under the context of cost-sensitive loss. Roughly speaking, it shows that if the two decision functions are dissimilar to each other, the fairness constraint will not harm too much on the target utility. As a comparison, our results (Theorem 3.1 and 3.2) directly give lower bounds on the sum of errors across groups in terms of the difference in base rates as well as the distance of representations. Our results are also different from those in Madras and Zhang et al.: they gave an upper bound on the demographic parity gap in terms of the loss incurred by an adversary (Theorem 5.1), while ours are about lower bounds on the errors of the target task.

**[Reviewer # 1, Other questions]** We use the notation $\mathcal{P} \ll \mathcal{Q}$ to mean that distribution $\mathcal{P}$ is absolutely continuous w.r.t. distribution $\mathcal{Q}$, i.e., for any measurable event $E$, if $\mathcal{P}(E) > 0$, then we must have $\mathcal{Q}(E) > 0$ as well. The generator function of KL divergence is indeed $f(t) = t \log t$, and the generator function of the inverse KL divergence is $f(t) = -\log t$. Having identical joint distributions implies that the optimal decision functions are the same across groups, but not the other way around. We also add one more experimental result with $\lambda = 50.0$, and the result is listed as follows. Compared with the existing results in Table 2, we can see a consistent trend.

|  | $\text{Err}_{\mathcal{D}}$ | $\text{Err}_{\mathcal{D}_0} + \text{Err}_{\mathcal{D}_1}$ | $|\text{Err}_{\mathcal{D}_0} - \text{Err}_{\mathcal{D}_1}|$ | $d_{\text{TV}}(\mathcal{D}_0(\hat{Y}), \mathcal{D}_1(\hat{Y}))$ |
|---|---|---|---|---|
| AdvDebias, $\rho = 50.0$ | 0.201 | 0.360 | 0.112 | 0.028 |

**[Reviewer # 3]** We are happy that the reviewer found our paper to be interesting, theoretically sound and well-written. As stated in the last sentence of the conclusion section, our lower bound naturally implies an algorithm based on instance-reweighting to balance the base rates during fair representation learning. However, the detailed design, analysis and empirical validation of such an algorithm is beyond the scope of current paper. Given that nowadays there are more than tens paper on proposing new algorithms to achieve fairness every year, we believe it would be nice to have a theoretical paper with novel analysis techniques and results to study the fundamental limit of such algorithms. Although it is clear that fairness will compromise utility, before this paper it is still unknown to what extent will it, and how is it related to the difference in terms of base rates across groups. From this perspective, we believe our work is a timely paper that answers the above questions quantitatively. As pointed out by Reviewer 1, our analysis technique using Liese and Vadja lemma is novel and useful. This is of independent interest and we expect its applicability in a broader context.

**[Reviewer # 4]** We would like to thank Reviewer 4 for the encouraging comments. As explained in Theorem 2.1., Chouldechova and Kleinberg et al. mainly proved that positive rate parity and predictive value parity are in general incompatible. This is an impossibility result between two different notions of fairness. As a comparison, we mainly focus on trade-off between utility and fairness. Furthermore, our Theorem 3.1 is a quantitative result in the sense that it not only gives the impossibility statement when base rates are different, but also gives a lower bound on the error that will be incurred by *any* algorithm. Techniques based on instance-reweighting helps to decrease the difference in base rates, and hence we would expect it to help decrease the lower bound as well. This means that we would incur less drop of utility when learning fair representations. Our current technique does not extend to the definition of equal opportunity, and collecting additional data will not help.

[1]. The Cost of Fairness in Binary Classification. Menon and Williamson, FAT* 2018.

[2]. MINE: Mutual Information Neural Estimation. Belghazi, ICML 2018.

[Meta-Review · NeurIPS 2019]

This paper shows information-theoretic lower bounds characterizing fairness-utility trade-offs in representation learning. The work is interesting, novel and timely, and has the potential to inspire new research directions in fairness in ML.